# A Fast and Non-Destructive Prediction Model for Remaining Life of Rigid Pavement with or without Asphalt Overlay

**Xuan Hong [1], Weilin Tan [2], Chunlong Xiong [3,4,*] , Zhixiong Qiu [1], Jiangmiao Yu [3], Duanyi Wang [3], Xiaopeng Wei [3], Weixiong Li [3,4] and Zhaodong Wang [4]**

1 Guangdong Expressway Co., Ltd., Guangzhou 510100, China; xuanh@gdfreeway.com (X.H.); qiuzx@gdfreeway.com (Z.Q.)
2 Guangdong Kaiyang Expressway Co., Ltd., Jiangmen 529200, China; twlin@gdfreeway.com
3 School of Civil Engineering and Transportation, South China University of Technology, Guangzhou 510640, China; yujm@scut.edu.cn (J.Y.); tcdywang@scut.edu.cn (D.W.); 202121010608@mail.scut.edu.cn (X.W.); 201810101689@mail.scut.edu.cn (W.L.)
4 Guangzhou Xiaoning Roadway Engineering Technology Research Institute Co., Ltd., Guangzhou 510641, China; zhaodongwang@xndl3.wecom.work
* Correspondence: cthgclx@mail.scut.edu.cn

**Abstract:** Remaining life is an important indicator of pavement residual effective service time and is directly related to maintenance decision-making with limited funds. This paper proposes a fast and non-destructive model to predict the remaining life of rigid PCC (Portland cement concrete) pavement, with or without asphalt overlay. Firstly, a model was constructed according to the current Chinese design specifications for concrete pavement integrating an inverse design concept. Secondly, the prediction model was applied to three typical pavement sections with 1430, 1250 and 1000 slabs, respectively. Ground penetrating radar (GPR) was utilized to determine the geometric parameters in the predictive model and the physical state of the pavement. A falling weight detector (FWD) was utilized for determination of the mechanical parameters. A more reasonable equivalent elastic modulus of foundation was back-calculated instead of using the limited model in the design specification. Thirdly, the remaining life was predicted based on the current mechanical and geometric parameters. The distributions of the remaining life of the three pavement sections was statistically analyzed. Finally, a decision-making system to inform maintenance strategy was proposed based on the remaining life and the technical condition of each slab. The results showed that the relationship between the remaining life and the mechanical parameters, geometric parameters and the physical state of the pavement was highly consistent with engineering experience. The success rate of the prediction model was as high as 96%. The proposed fast and non-destructive prediction model showed good engineering applicability and feasibility. The decision-making system was shown to be feasible in terms of economic benefits.

**Keywords:** remaining life; rigid PCC pavement; GPR; FWD; inverse design concept

## 1. Introduction

### 1.1. Background

Portland cement concrete (PCC) pavement, with or without asphalt overlay, is widely used because of its good durability, stability, strong bearing capacity, long service life, and low maintenance cost [1,2]. In recent years, with the rapid development of the economy, the amount of traffic has increased sharply, and axle loads have become increasingly heavy. PCC pavements usually show serious performance degradation before they reach the intended service life [3,4]. The concept of the remaining life of a pavement is based on the premise of directly repairing particular road damage or designing an asphalt overlay to cover the original surface defects [5,6]. Remaining life represents the rest of valid service life until a failure is reached in the PCC pavement as a result of exposure to traffic and

environmental forces [7]. The concept of remaining life is helpful for making optimal use of the capacity for in-service pavement repair and facilitating decision-making for reconstruction or rehabilitation with limited existing resources [8,9].

The remaining life of a PCC pavement can be understood in terms of two categories: functional and structural [10,11]. The functional category relates to surface problems and mainly relies on pavement surface indicators that can be directly detected. The most representative method is the serviceability approach involving the combining of performance prediction equations and triggering thresholds for maintenance or rehabilitation [12–14]. In contrast, structural remaining life relates to internal problems and mechanical indicators that cannot be detected directly [15]. Currently, there is no recognized approach to the verification of structural remaining life. Prediction methods for the assessment of the in-site structural remaining life of PCC pavement have developed little over recent decades. The most representative method is the 1993 AASHTO NDT approach using field test mechanical data [16]. Another structural remaining life (fatigue life) prediction approach is based on laboratory-measured material mechanical degradation [17]. Additionally, statistical approaches, based on quite large amounts of data (especially concerning maintenance implementation time) are used to directly calculate the service life of a particular PCC pavement [18]. The existing methods for predicting the remaining life of rigid PCC pavement utilize various concepts ranging from the purely empirical to the entirely mechanistic. However, it has always been a difficult and complex problem to accurately predict the remaining life of rigid PCC pavement.

A review of the literature indicates that surface functional failure does not have a simple relationship with pavement inner condition [19–23]. The predicted functional remaining life of an in-service pavement with a recent new overlay may be seriously overestimated. The new overlay may be prematurely destroyed before the functional remaining life is reached. Consideration of the structural remaining life is more reasonable and scientific. Firstly, with respect to the 1993 AASHTO NDT approach, this requires details of the effective thickness, determined by modulus back-calculation, and the original slab thickness, using an empirical graphic procedure. Unfortunately, the effective thickness is subjectively assessed and is arguably unreliable [24]. The modulus back-calculation method used is based on a flexible pavement which, theoretically, is not applicable to rigid PCC pavement [25]. Secondly, the mechanical degradation approach, which is based on a fatigue criterion for concrete materials, does not consider the combination of structure and thickness [26,27]. Lastly, the statistical approach is the most direct and most closely related to the field situation, but has considerable drawbacks, including that the time taken for statistical analysis is extensive, and the failure time is difficult to accurately assess. Moreover, the regular or periodic road maintenance occurring will have a significant impact on the remaining life of the pavement. For different projects, there will also be environmental and traffic differences. Thus, statistically based prediction of the remaining life of a particular pavement might be limited to this situation and be purely empirically based.

To address the shortcomings discussed above, a methodology which incorporates an inverse design concept for prediction of the remaining life of a rigid PCC pavement was proposed. Three typical pavement sections were selected for application of the prediction model. Non-destructive detecting technology, including GPR and FWD, was utilized to obtain data on pavement technical condition.

### 1.2. Objective and Scope

The motivation for the study was the development of a remaining life prediction model of rigid PCC pavement, with or without asphalt overlay, applying integrated non-destructive detection technologies to provide details of on-site pavement parameters to improve the speed of prediction, accuracy, and representativeness.

## 2. Concept and Methodology Development

### 2.1. Inverse Design Concept

In this study, a methodology for predicting the remaining life of rigid PCC pavement is proposed which is based mainly on an inverse design concept, in contrast to the traditional forward design procedure described in the JTG D40-2011 Specifications for Design of Highway Cement Concrete Pavement [28]. The forward design approach determines the road parameters, including road materials and thickness combinations, according to the assumed number of axles load and required pavement conditions. However, the inverse design concept involves evaluation of pavement condition based on the current state of pavement materials, structure thickness combinations, and current and future traffic trends [29].

### 2.2. Methodology Development

#### 2.2.1. Remaining Life Prediction Model

Combining the current state and the predicted future traffic volume, the remaining service life of a PCC pavement is calculated using an inverse design concept as described below.

In the forward design, the cumulative number of axle loads during the design period, $N_e$, is calculated using Equation (1) according to measured traffic parameters, such as the initial traffic volume and the traffic volume growth rate [28].

$$N_e = \frac{N_1\left((1+\gamma)^t - 1\right) \times 365}{\gamma} \tag{1}$$

where, $\gamma$ is the annual average growth rate of traffic over the design life. The initial number of axle loads, $N_1$, is determined based on traffic volume survey and calculated according to Equations (2) and (3) [30].

$$N_1 = AADTT \times DDF \times LDF \times \sum_{m=2}^{11} (VCDF_m \times EALF_m) \tag{2}$$

$$EALF_m = \sum_i \left(\frac{NA_{mi}}{NT_m} \sum_j \left(c_1 c_2 \left(\frac{P_{mij}}{P_s}\right)^{16} \times \frac{ND_{mij}}{NA_{mi}}\right)\right) \tag{3}$$

where, $AADTT$ is the two-way annual average daily traffic volume of 2-axle 6-wheel and above vehicles; $DDF$ is a direction factor determined based on the ratio of the number of vehicles in the two directions; $LDF$ is a lane coefficient determined according to the ratio of the number of vehicles in the design lane; $VCDF_m$ is the type distribution coefficient of $m$-class vehicles; $EALF_m$ is the conversion factor of the equivalent design axle load for $m$-class vehicles; $NA_{mi}$ is the total number of $i$ axle types in m-class vehicles; $NT_m$ is the total number of $m$-class vehicles; $P_{mij}$ is the single-axle axle load of $i$-type axle in $j$-level axle load range of $m$-class vehicles; $P_s$ is the uniaxial weight of design axle load ( usually 100 kN in China); and $ND_{mij}$ is the number of $i$-type axles in the $j$-level axle load range in $m$-class vehicles. Factor $c_1$ is the axle group factor—1 for a single axis, 2.6 for a double shaft and 3.8 for a triple shaft—based on engineering experience in China. Factor $c_2$ is the wheel coefficient—1 for a double wheel and 4.5 for a single wheel.

Therefore, the design life, $t$, can be calculated and deduced as Equation (4) based on Equations (1)–(3).

$$t = \log\left(\frac{N_e}{365 N_1}\gamma + 1\right) / \log(\gamma + 1) \tag{4}$$

The design life is a set target value. The general forward design procedure involves the following steps: (1) assumes the initial pavement parameters, such as thickness, modulus, strength; (2) statistically predict the cumulative number of axial loads in the target design life period; (3) input the pavement parameters and the cumulative number of axial loads into the mechanical model; (4) calculate the load-temperature coupling stress of the bottom

of the cement concrete slab; and (5) adjust the initial pavement parameters until the load-temperature coupling stress is less than the flexural tensile strength of the designed cement concrete. The finalized pavement parameters are the design results that can meet the goal of the design life.

From the perspective of inverse design, where $t$ is regarded as the remaining life of the pavement, $N_e$ should be no larger than the cumulative number of axial loads predicted based on Equation (1), but the number of axial loads that the current pavement can still bear, which is directly related to the current pavement parameters.

In the general forward design procedure, $k_f$ is the fatigue stress coefficient considering the cumulative fatigue effect of load stress during the design reference period, which represents the pavement parameters and has a relationship with the number of axial loads that the current pavement can still withstand. Their relationship can be expressed as Equation (5) [28]:

$$k_f = N_e^\theta \tag{5}$$

In Equation (5), the value of $\theta$ is generally 0.057 when Portland cement concrete is used in the surface layer and 0.065 for the base layer. In the prediction model of the remaining life of rigid PCC pavement, $t$, can be deduced and expressed as Equation (6).

$$t = \log\left(\frac{k_f^{1/\theta}}{365N_1}\gamma + 1\right)/\log(\gamma + 1) \tag{6}$$

In Equation (6), the current number of axle loads, $N_1$, can be calculated as Equation (2), and the annual average growth rate of traffic can be calculated based on the historical traffic data. $k_f$ is a mechanical index relating to the current pavement parameters, such as thickness, modulus and strength and the cumulative fatigue effect of load-temperature coupling stress.

### 2.2.2. Determination of Model Parameters

In order to obtain $k_f$ for the current pavement structure, it is necessary to carry out the analysis from the perspective of pavement mechanics calculations. In Equation (6), $k_f$ is calculated as Equation (7) [28].

$$k_f = \sigma_{pr}/(\sigma_{ps}k_ck_r) \tag{7}$$

where, $k_c$ is the compressive coefficient (usually 1.15 for highway and 1.10 for first class highway); $k_r$ is the stress reduction factor (usually 0.85); and $\sigma_{ps}$ is the stress generated by the design axle load at the critical position of a four-side free plate calculated as Equations (8)–(10) [28].

$$\sigma_{ps} = 1.47 \times 10^{-3} r^{0.70} h_c^{-2} P_s^{0.94} \tag{8}$$

where, $P_s$ is the uniaxial weight of the design axle load (usually 100 kN); and $r$ is the relative stiffness radius of the concrete layer. The factor, $h_c$, is the concrete layer thickness [28].

$$r = 1.21 \times (D_c/E_t)^{1/3} \tag{9}$$

where, $E_t$ is the equivalent elastic modulus of the foundation underneath the concrete layer.; and $D_c$ is the relative stiffness radius of the concrete layer [28].

$$D_c = \frac{E_c h_c^3}{12(1 - v_c^2)} \tag{10}$$

where, $E_c$ is the concrete layer bending modulus; $v_c$ is the Poisson ratio of concrete, which is a constant during the elastic deformation phase of the material, and $v_c$ is usually 0.15 of the cement concrete.

The driving load fatigue stress generated at the critical load position in a rigid concrete slab, $\sigma_{pr}$ in Equation (7), is deduced and calculated as Equation (11). It reflects the

mechanical equilibrium state of the pavement; that is, the flexural tensile strength of the cement concrete is exactly equal to the load-temperature coupling stress of the bottom of the rigid PCC pavement.

$$\sigma_{pr} = \frac{f_r}{\gamma_r} - \sigma_{tr} \tag{11}$$

where, $f_r$ is the standard value of the bending strength of cement concrete. The design value is usually 5 MPa but the on-site measured value is usually different from the design value. $f_r$ can also be deduced and calculated as Equation (12). $\gamma_r$ is the reliability coefficient related to the target reliability, variation level and variation coefficient (usually 1.64 for highway and 1.28 for first-class highway).

$$f_r = \frac{0.96}{\frac{10^4}{E_c} - 0.09} \tag{12}$$

$\sigma_{tr}$ is the driving load fatigue stress generated at the critical load position in a rigid concrete slab calculated as Equation (13) [28].

$$\sigma_{tr} = k_t \sigma_{t,\max} \tag{13}$$

where, $k_t$ is the temperature fatigue stress coefficient calculated as Equation (14) [28].

$$k_t = \frac{f_r}{\sigma_{t,\max}} \left( a_t \left( \frac{\sigma_{t,\max}}{f_r} \right)^{b_t} - c_t \right) \tag{14}$$

where, $a_t$, $b_t$, $c_t$ are the model parameters considering the natural environment in which the road is located. For roads in Guangdong province in China, $a_t$ is usually 0.841, $b_t$ is usually 1.323, and $c_t$ is usually 0.058. $\sigma_{t,\max}$ is the maximum temperature stress of a concrete slab under a maximum temperature gradient calculated as Equation (15) [28].

$$\sigma_{t,\max} = \frac{\alpha_c E_c h_c T_g \zeta}{2} B_L \tag{15}$$

where, $\alpha_c$ is the concrete linear expansion coefficient (for Portland cement concrete the coefficient is usually $1.1 \times 10^{-5}/°C$); and $T_g$ is the 50-year maximum temperature gradient at the location of the road. For roads in Guangdong province, the statistical result of $T_g$ is 88 °C/m. The factor, $\zeta$, the correction factor of the temperature gradient, can be obtained by regression analysis as Equation (16), according to the current specification. Its correlation coefficient value can be as high as 0.99.

$$\zeta = 18.56 h_a^2 - 8.58 h_a + 1.29 \tag{16}$$

$B_L$ is the temperature stress coefficient of integrated temperature warping stress and internal stress calculated as Equations (17)–(19) [28].

$$B_L = 1.77 e^{-4.48 h_c} C_L - 0.131 (1 - C_L) \tag{17}$$

$$C_L = 1 - \frac{\sinh\delta \cos\delta + \cosh\delta \sin\delta}{\cos\delta \sin\delta + \sinh\delta \cosh\delta} \tag{18}$$

$$\delta = \frac{L}{3r} \tag{19}$$

where, $C_L$ is the temperature warping stress coefficient of the concrete slab. Factor $L$ is the length of the slab.

When the rigid PCC pavement is covered with the asphalt overlay, the remaining life prediction model may be different from the prediction model in Equation (6). It must consider the effect of the overlay on the concrete layer's load-induced fatigue stress and

temperature-induced fatigue stress. Equations (8) and (13) are adjusted by Formulas (20) and (21).

$$\sigma_{psa} = (1 - \delta_a h_a)\sigma_{ps} \tag{20}$$

$$\sigma_{tra} = (1 + \delta_a' h_a)\sigma_{tr} \tag{21}$$

where, $\sigma_{psa}$ is the load-induced stress generated by the axle load at the critical position of the concrete slab with the asphalt overlay; factor $\sigma_{tra}$ is the temperature-induced stress generated by the axle load at the critical position of the concrete slab with asphalt overlay; factor $h_a$ is the thickness of the overlay; and $\delta_a$ and $\delta_a'$ are the model factors obtained by regression analysis as Equations (22) and (23), based on the current specification. Their correlation coefficients are 0.986 and 0.972, respectively.

$$\delta_a = 2.69 - 0.001\frac{E_c}{E_t} - 3.77h_c \tag{22}$$

$$\delta_a' = 1.98 - 0.018 \times 10^{-3}E_c - 2.94h_c \tag{23}$$

In sum, Equations (7)–(23) describe in detail the acquisition or calculation method for each parameter in the remaining life of the rigid PCC pavement prediction model in Equation (6). Some of the parameters that are determined by experience have been described in the above text; some of the other parameters that need to be obtained are classified in Table 1.

**Table 1.** Model parameters needing to be determined.

| Classification | Parameters | Expressions | Methods in This Research |
|---|---|---|---|
| Traffic data | $AADDT$ | two-way annual average daily traffic volume of 2-axle and 6-wheel and above vehicles | Measured by imaging, weighing and other equipment installed on site by the management department |
| | $DDF$ | direction factor | |
| | $LDF$ | lane coefficient | |
| | $VCDF_m$ | type distribution coefficient of $m$-class vehicles | |
| | $NA_{mi}$ | total number of $i$ axle types in $m$-class vehicles | |
| | $NT_m$ | total number of $m$-class vehicles | |
| | $ND_{mij}$ | number of $i$-type axle in the $j$-level axle load range in $m$-class vehicles | |
| | $P_{mij}$ | single-axle axle load of $i$-type axle in $j$-level axle load range of $m$-class vehicles | |
| | $\gamma$ | annual average growth rate of truck traffic during the reference period | Calculated based on historical traffic data |
| Geometry | $h_a$, $h_c$, $L$ | thickness of asphalt overlay and concrete slab, length of slab | Fast detection by radar |
| Mechanical parameter | $E_c$ | concrete layer bending modulus | Detected by falling weight deflector and backcalculated modulus |
| | $E_t$ | equivalent elastic modulus of foundation underneath the concrete layer | |

In Table 1, the traffic data can be obtained by data survey with the help of the local transportation management department. The geometry and the mechanical parameters of the rigid PCC pavement can be obtained by replacing rapid and non-destructive field tests, such as GPR and FWD, for traditional single-point core drilling and laboratory testing. The total number of geometry and mechanical parameters is only five for rigid pavement with asphalt overlay and only four for rigid pavement without asphalt overlay, which greatly simplifies the life prediction model in Equation (6).

### 3. Field Testing

*3.1. Pavement Sections*

Three rigid pavement sections, with or without asphalt overlay of the same length, from the Guangdong Province in China, were selected as study objects. Basic information for the three pavements is shown in Table 2.

**Table 2.** Basic information for the pavement sections.

| Section | S118 TP | S118 HO | XL |
|---|---|---|---|
| Type | Rigid | Rigid with asphalt overlay | Rigid |
| Grade | First class | First class | Highway |
| Length/km | 5 | 5 | 5 |
| Age/a | 26 | 27 | 22 |
| Structure combination | Concrete layer 28 cm + cement stabilized base 24 cm + subgrade | Asphalt overlay 3 cm + concrete layer 25 cm + gravel layer 50 cm + subgrade | Concrete layer 28 cm + limestone stabilized base 20 cm + gravel layer 20 cm + subgrade |

*3.2. Data Collection and Process*

3.2.1. Traffic Data

Traffic data was surveyed with the help of the local transportation management department, to convert the vehicle loads of different axle loads of different axle types of different vehicle types into standard design axle loads. At present, some mature commercial websites can be used to convert traffic axle loads with the above basic survey data, such as goodpave.com (accessed on 20 February 2022) [31] and daokedaopave.com (accessed on 25 February 2022) [32].

3.2.2. Geometrical Data

Three-dimensional(3D) ground penetrating radar (GPR) was used to detect the pavement layer thickness and slab length. The radar was produced by KONTUR, the radar host was GEOSCOPE MK IV, and the ground-coupled antenna was type DXG1820 with a frequency bandwidth of 200–3000 MHz. GPR was used to find physical defects, such as voids, broken slabs, uneven settlement of subgrades, etc. Measurements were taken continuously at speeds ranging from 20 km/h to 30 km/h. Trigger spacing, time window and dwell time were set separately at 2.5 cm, 25 ns and 3 us, respectively. The survey width was no more than 1.5 m, with each lane assessed three times for full-width inspection [33]. Approximately 15 km was inspected separately for each of the sections S118-TP, S118-HO and XL. The principle and components of GPR are shown in Figure 1a. Inspection using GPR is shown in Figure 1b.

3.2.3. Mechanical Parameters

FWD was applied to determine the concrete layer bending modulus and the equivalent elastic modulus of the foundation underneath the concrete layer using advanced and proven modulus back-calculation technology. The FWD was produced by Grontmij Carl Bro; the model used was Phonix PRI2100. The FWD inspection speed was approximately 2~3 km/h. An impulse load of $100 \pm 2.5$ kN was applied in the center area of each PCC slab [34]. For rigid pavement with an asphalt overlay, the boundary of the slab can first be determined by 3D GPR and an RTK (real-time kinematic) positioning system. There were about 1430, 1250 and 1000 test points separately for S118-TP, S118-HO and XL. For each point, deflections, air temperature, road surface temperature, and chainage were recorded. The technical principle of FWD and the inspection process are shown in Figure 2.

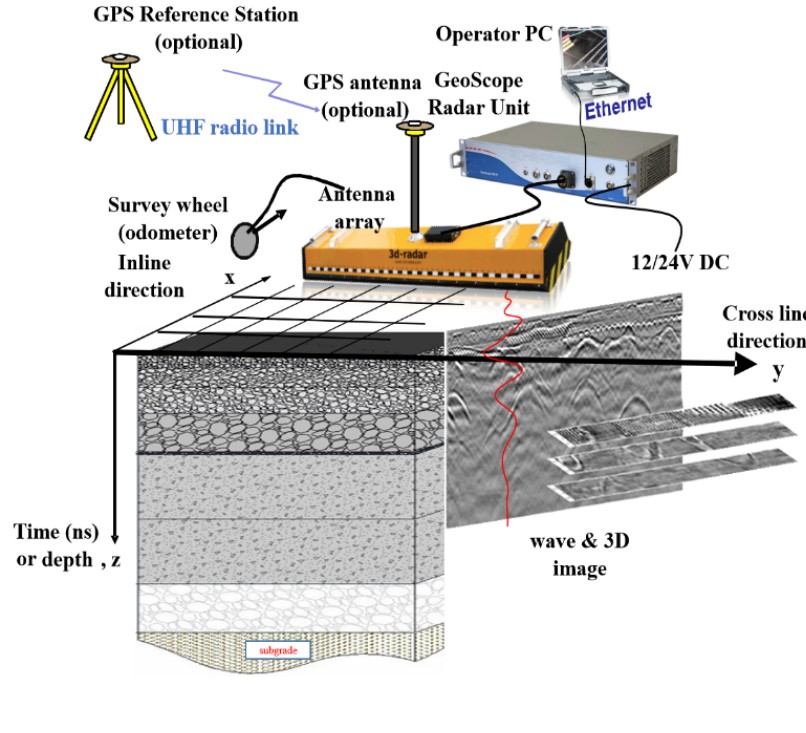

(**a**)

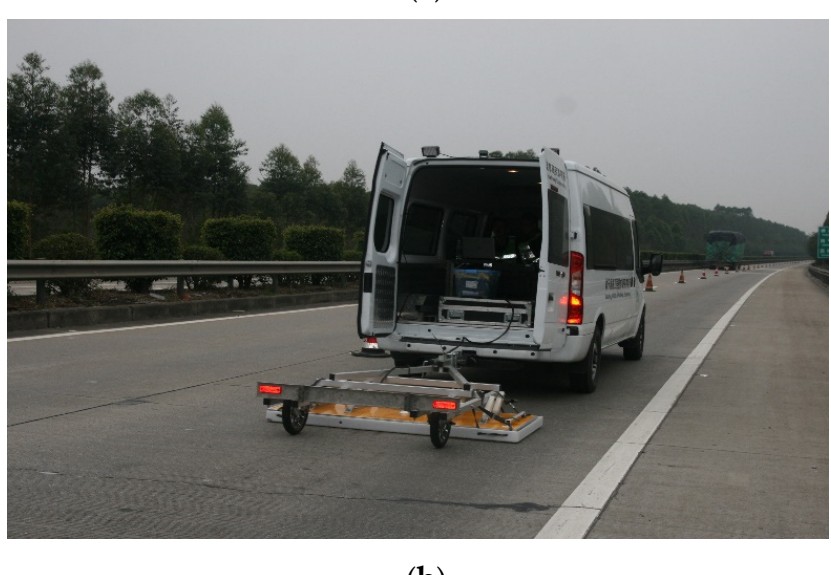

(**b**)

**Figure 1.** Structure condition and thickness inspection: (**a**) Principle of GPR; (**b**) On-site inspection using GPR.

To reduce dynamic fluctuation effects, it is necessary to ensure that there are no heavy vehicles passing when the FWD is working. The road surface should be smooth and free from debris to reduce deviation of the sensors. Although temperature and humidity have little influence on the modulus of a rigid PCC pavement, the temperature change range was set to be less than 3 °C, and the humidity changed as little as possible. The concrete layer bending modulus and the equivalent elastic modulus of foundation were both back-calculated using software reported in previous research [35].

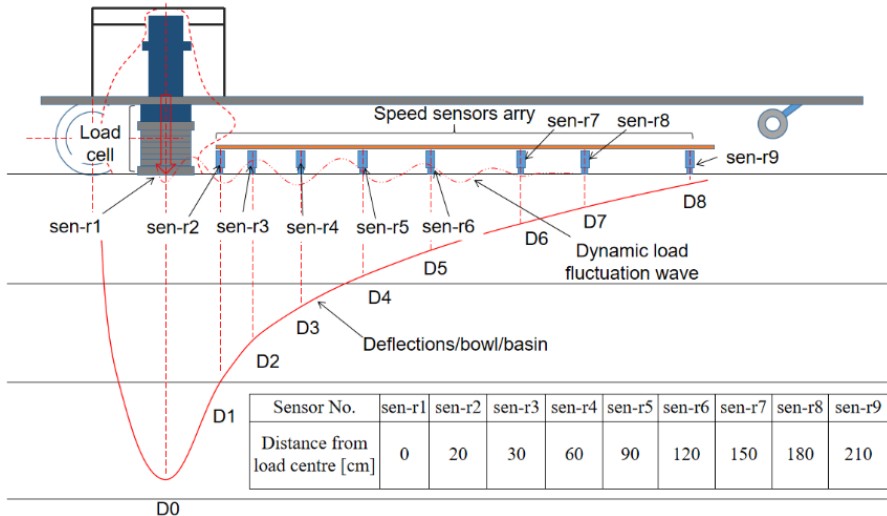

**(a)**

**(b)**

**Figure 2.** Inspection of pavement deflection basin: (**a**) technical principle of FWD; (**b**) FWD inspection.

## 4. Results and Analysis

### 4.1. Traffic Data

The two-way annual average daily traffic volume of 2-axle and 6-wheel and above vehicles ($AADDT$), the proportion of vehicles in two driving directions ($DDF$) and the proportion of vehicles in each lane in each direction ($LDF$) are shown in Table 3. The proportion of 2–11 types of vehicles ($VCDF_m$) is shown in Table 4 and the proportion of vehicles with each axle type in each type of vehicle (ratio of $NA_{mi}$ and $NT_m$) is shown in Table 5. The proportion of vehicles in different axle load weight ranges in each axle-type vehicle (ratio of $ND_{mij}$ and $NA_{mi}$) is the axle weight distribution coefficient. The axle weight distribution coefficient of single-axle single-tire of class 2–11 vehicles of S118 TP is shown in Figure 3.

**Table 3.** General traffic parameters.

| Parameters | S118 TP | S118 HO | XL |
|---|---|---|---|
| *AADDT* | 25,780 | 16,593 | 48,956 |
| *DDF* | 0.5 | 0.6 | 0.55 |
| *LDF* | 1 | 1 | 0.7 |
| $\gamma$ | 1.5% | 1.2% | 7.0% |

**Table 4.** Proportion of vehicle types 2–11 ($VCDF_m$).

| Type of Vehicle | S118 TP | S118 HO | XL |
|---|---|---|---|
| 2 | 17.80 | 28.90 | 22.00 |
| 3 | 33.00 | 43.80 | 23.30 |
| 4 | 3.40 | 5.50 | 2.70 |
| 5 | 0.00 | 0.00 | 0.00 |
| 6 | 12.50 | 9.40 | 8.30 |
| 7 | 4.40 | 2.00 | 7.50 |
| 8 | 9.10 | 4.60 | 17.10 |
| 9 | 10.60 | 3.40 | 8.50 |
| 10 | 8.50 | 2.30 | 10.60 |
| 11 | 0.70 | 0.10 | 0.00 |

**Table 5.** Proportion of vehicles with each axle type in each type of vehicle ($NA_{mi}/NT_m$).

| Type of Vehicle | S118 TP | | | | S118 HO | | | | XL | | | |
|---|---|---|---|---|---|---|---|---|---|---|---|---|
| | Single Axle Single Tire | Single Axle Double Tire | Double Shaft | Triple Shaft | Single Axle Single Tire | Single Axle Double Tire | Double Shaft | Triple Shaft | Single Axle Single Tire | Single Axle Double Tire | Double Shaft | Triple Shaft |
| 2 | 1.00 | 1.00 | 0.00 | 0.00 | 1.00 | 1.00 | 0.00 | 0.00 | 1.00 | 0.99 | 0.01 | 0.00 |
| 3 | 1.00 | 1.00 | 0.00 | 0.00 | 1.00 | 1.00 | 0.00 | 0.00 | 1.00 | 1.00 | 0.00 | 0.00 |
| 4 | 1.00 | 0.00 | 1.00 | 0.00 | 1.00 | 0.00 | 1.00 | 0.00 | 1.00 | 0.00 | 1.00 | 0.00 |
| 5 | 1.00 | 0.00 | 0.00 | 1.00 | 1.00 | 0.00 | 0.00 | 1.00 | 1.00 | 0.00 | 0.00 | 1.00 |
| 6 | 2.00 | 0.38 | 0.62 | 0.00 | 2.00 | 0.43 | 0.57 | 0.00 | 2.00 | 0.50 | 0.50 | 0.00 |
| 7 | 1.00 | 1.00 | 1.00 | 0.00 | 1.00 | 1.00 | 1.00 | 0.00 | 1.00 | 1.00 | 1.00 | 0.00 |
| 8 | 1.00 | 0.56 | 0.89 | 0.56 | 1.00 | 1.00 | 0.00 | 1.00 | 1.00 | 0.93 | 0.14 | 0.93 |
| 9 | 1.00 | 0.00 | 1.00 | 1.00 | 1.00 | 0.00 | 1.00 | 1.00 | 1.00 | 0.00 | 1.00 | 1.00 |
| 10 | 2.00 | 1.00 | 0.04 | 0.96 | 2.00 | 1.00 | 0.09 | 0.91 | 2.00 | 1.00 | 0.15 | 0.85 |
| 11 | 0.00 | 0.00 | 0.00 | 0.00 | 0.00 | 0.00 | 0.00 | 0.00 | 0.00 | 0.00 | 0.00 | 0.00 |

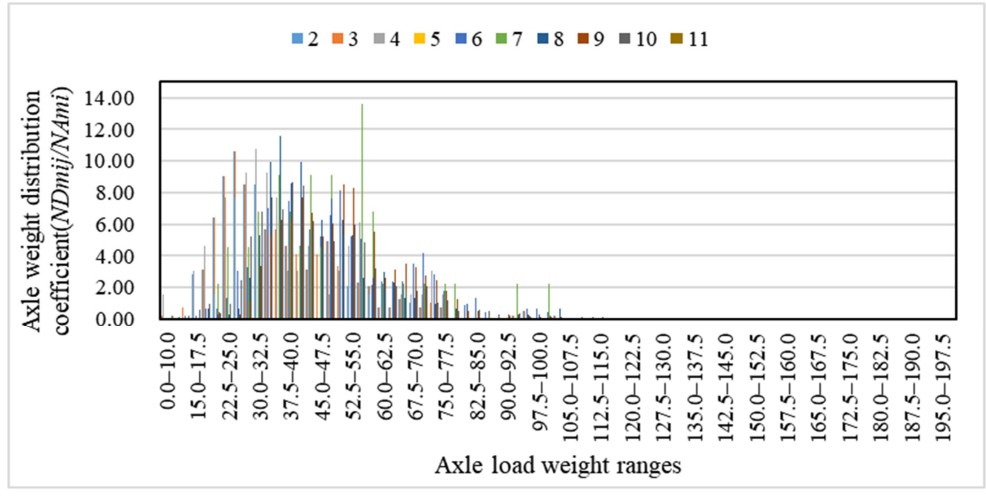

**Figure 3.** Axle weight distribution coefficient of single-axle single-tire of class 2–11 vehicles of S118 TP.

Based on the above survey data, $N_1$ of S118 TP, S118 HO, and XL were calculated as 12,890, 9956 and 18,848, respectively using Equations (2) and (3) with the assistance of the daokedaopave.com website (accessed on 25 February 2022).

### 4.2. Geometrical Data

From the GPR post-processed data, the number of slabs in the design lane of S118 TP, S118 HO, and XL were 1430, 1250 and 1000, respectively. The mean length of the slabs of S118 TP, S118 HO, and XL were 3.5 m, 4.0 m and 5.0 m, respectively. Their coefficients of variation were 1%, 2% and 0%, respectively. The length of slabs on the same section fluctuated normally, but the values were relatively consistent. The slab length distributions of the design lanes of S118 TP, S118 HO, and XL are shown in Figure 4.

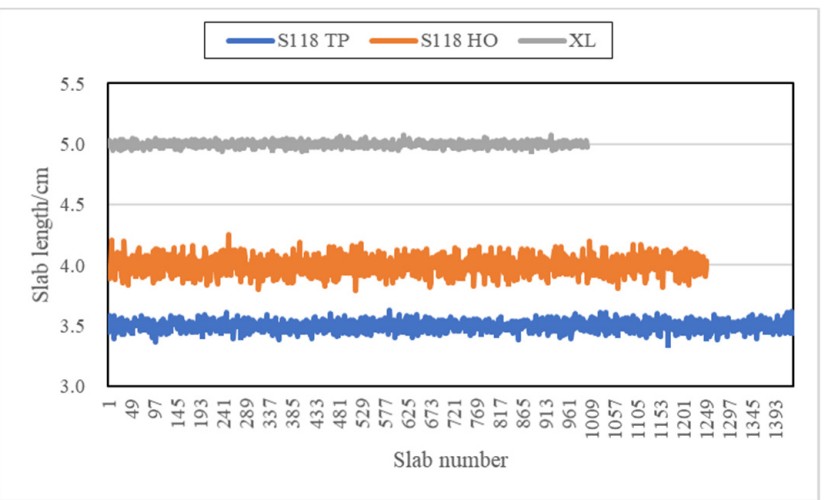

**Figure 4.** Slab length distribution of the design lanes of S118 TP, S118 HO, and XL.

The mean thickness of the slabs of S118 TP, S118 HO, and XL were 26.9 cm, 24.3 cm and 26.3 cm, respectively Their coefficients of variation were 8%, 6% and 3%, respectively Differences in the thickness of the slabs for the same pavement section were not obvious. The slab thickness distributions of the design lanes of S118 TP, S118 HO, and XL are shown in Figure 5. The geometrical data for slabs were affected by the construction error control standards of different highway grades.

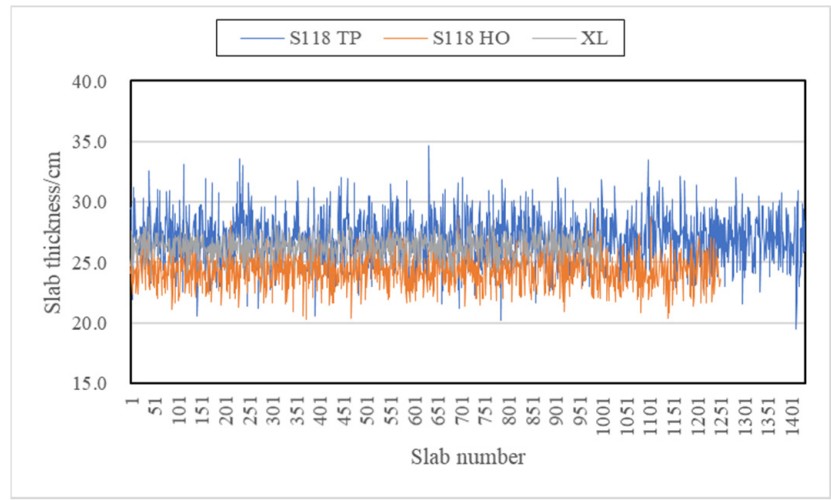

**Figure 5.** Slab thickness distribution of the design lanes of S118 TP, S118 HO, and XL.

The mean thickness of the asphalt overlay of S118 HO was 3.0 cm and its coefficient of variation was 36%. The asphalt overlay thickness distribution of the design lane of S118 HO is shown in Figure 6. The thickness of the asphalt overlay was related to the flatness of the cement slab surface, the construction quality control, and the compression deformation caused by the vehicle load.

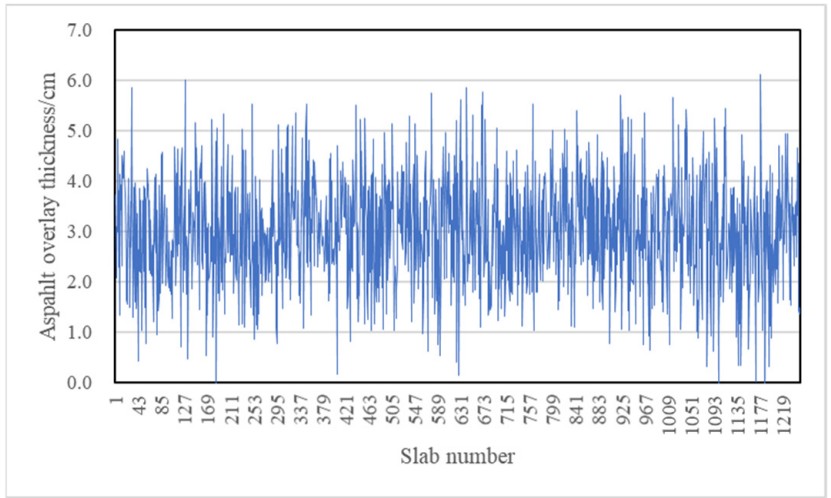

**Figure 6.** Asphalt overlay thickness distribution of the design lane of S118 HO.

### 4.3. Mechanical Parameters

The equivalent elastic modulus of the foundation underneath the concrete layer is an important indicator for PCC pavement forward design [36] and is also a factor for predicting rest of life during inverse design. The current specification provides a model for calculating the equivalent elastic modulus of the foundation based on the deflections. However, the application scope of the model is limited to the cement concrete pavement without asphalt overlay. The self-weight of the asphalt overlay and the interlayer bonding between the slab and the overlay will limit the bending deformation of the slab under load and reduce the measured deflection value. The model will overestimate the equivalent elastic modulus of foundation by about 8% based on the actual data.

To obtain a more reasonable equivalent elastic modulus of foundation, a back-calculation model is constructed. Two- or three-layer mechanical models in the back-calculation process for rigid pavement, with or without asphalt overlay, were adapted from the multilayer (more than two or three layers) mechanical model shown in Figure 7. Through this model, the equivalent elastic modulus of foundation, the concrete layer bending modulus, and the modulus of asphalt overlay can be obtained at the same time.

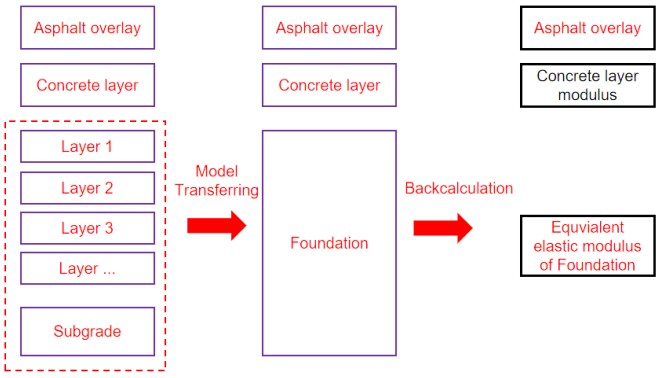

**Figure 7.** Model transferring for the back-calculation method.

The equivalent elastic moduli for the foundation distributions of the design lane of S118 TP, S118 HO and XL are shown in Figure 8. The mean of XL was 423 MPa, which was the largest among the three sections and its coefficient of variation was about 15%, representing the smallest. The mean of S118 TP was relatively larger than that of S118 HO but it had the largest variability of about 41%. The bearing capacity of the structure underneath the concrete layer of XL was the highest and its uniformity was best, that of S118 HO was the lowest but its uniformity was better than the uniformity of S118 TP.

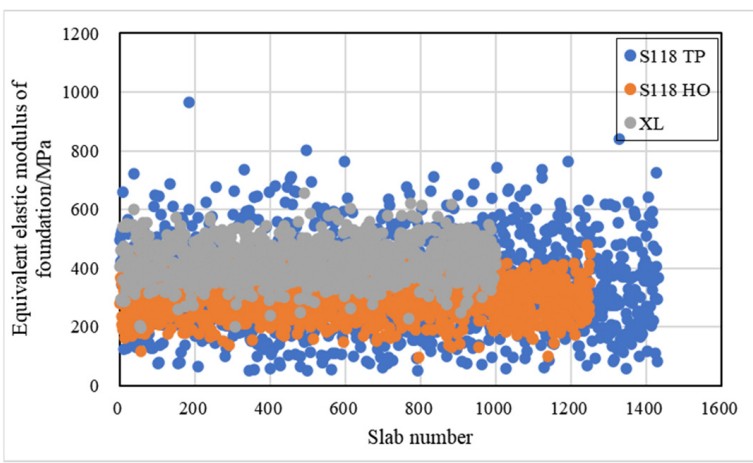

**Figure 8.** Equivalent elastic modulus of foundation distribution for the design lanes of S118 TP, S118 HO and XL.

The concrete layer bending modulus distributions of the design lanes of S118 TP, S118 HO and XL is shown in Figure 9. The concrete layer bending modulus of S118 TP ranged from 12,187 MPa to 61,455 MPa with a mean value of 31,334 MPa; the variability was about 17.8%. That of S118 HO ranged from 10,659 MPa to 48,025 MPa with a mean value of 29,853 MPa with a variability of about 19.3%. That of XL ranged from 16,385 MPa to 61,214 MPa with a mean value of 37,685 MPa with a variability of about 20.8%. The data showed that XL had the largest mean concrete layer bending modulus, but also had a relatively large coefficient of variation. The concrete layer bending modulus results for S118 TP and S118 HO were relatively less biased, but S118 TP had the smallest variability. These phenomena may have resulted from the extent of fine construction and quality management of pavements depending on the highway grade and the traffic volume.

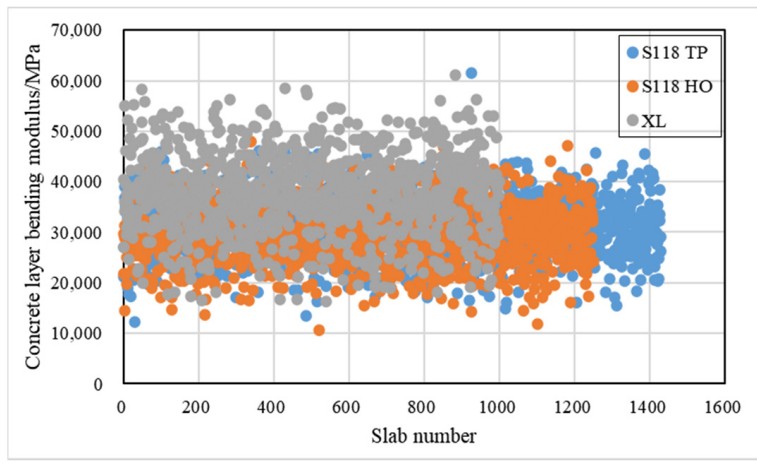

**Figure 9.** Concrete layer bending modulus distribution for the design lanes of S118 TP, S118 HO and XL.

In order to verify the accuracy of the back-calculated results, the modulus of the drilling cylindrical specimen of slabs was tested by conducting splitting bending-tension

tests indoors [24]. From the splitting bending-tension test results of 30 random drilling core samples in different pavement sections, the ratio of the back-calculated modulus and the test bending modulus ranged from 0.9 to 1.1 with a mean ratio of 1.01. The back-calculated results were very close to the results obtained in the laboratory.

### 4.4. Feasibility of the Prediction Model

A simple program based on the proposed prediction model was independently developed using Microsoft office software, and the remaining life of the pavement structure corresponding to each slab in the three pavement sections was calculated.

Based on the prediction results, the number of slabs of the 1430 slabs of S118 TP which could not be predicted was only 47, all 1250 slabs of S118 HO could be predicted, and there were only 15 slabs which could not be predicted of the 1000 slabs of XL. The success rate of the prediction model when applied to the three pavement sections was as high as 96%.

To verify the reliability of the predicted remaining life results, the technical condition characteristics of the pavement sections in the different remaining life intervals were classified; the results are shown in Table 6. The bending modulus of slab $E_c$ was significantly positively related to the remaining life and the correlation coefficients were all over 0.7. The thickness of the slabs of S118 TP and XL were significantly positively related to the remaining life and the correlation coefficients were over 0.8. The thickness of the slab of S118 HO was also positively related to the remaining life; however, the correlation coefficient was less than 0.5. The thickness of the asphalt overlay was weakly positively related to the remaining life. The equivalent elastic modulus of foundation $E_t$ of S118 TP and XL was weakly negatively related to the remaining life; however, the equivalent elastic modulus of the foundation of S118 HO was weakly positively correlated with the remaining life.

**Table 6.** Technical condition characteristics of the pavement in different remaining life intervals.

| Intervals | $E_c$ | | | $E_t$ | | | $h_c$ | | | $h_a$ | Physical State |
|---|---|---|---|---|---|---|---|---|---|---|---|
| | S118 TP | S118 HO | XL | S118 TP | S118 HO | XL | S118 TP | S118 HO | XL | S118 HO | |
| Less than 0.1a | 27,430 | 27,465 | 36,813 | 377 | 296 | 424 | 26.0 | 24.1 | 26.3 | 2.9 | Obvious physical defects such as voids, broken slabs, severe reflective cracks in overlay and pumping, etc. |
| 0.1a–1a | 32,006 | 33,873 | 49,915 | 338 | 296 | 414 | 26.9 | 24.7 | 26.5 | 3.3 | Slight physical defects such as voids, broken slabs, severe reflective cracks in overlay and pumping, etc. |
| 1a–5a | 32,992 | 35,688 | 51,498 | 314 | 305 | 407 | 27.3 | 24.9 | 26.6 | 3.5 | Uneven subgrade settlement under the slab, reflective cracks in overlay, etc. |
| 5a–10a | 34,455 | 37,440 | 52,933 | 327 | 305 | 384 | 27.2 | 24.8 | 26.3 | 3.2 | Slight reflective cracks in overlay, etc. |
| 10a–15a | 33,549 | 37,676 | 53,557 | 309 | 309 | 369 | 28.0 | 25.0 | 26.5 | 3.4 | Intact |
| More than 15a | 37,190 | 39,961 | 55,540 | 321 | 295 | 429 | 28.3 | 25.3 | 27.0 | 3.3 | Intact |

The results indicate that the statistical remaining life intervals had a high level of correspondence with the physical state of the pavement. The lower the predicted remaining life, the worse the corresponding technical condition of the pavement. This suggests that the prediction model proposed in this research has good engineering applicability and feasibility.

### 4.5. Application of the Prediction Model

The distributions of predicted remaining life of the three pavement sections for different intervals is shown in Figure 10. Most of slabs of S118 HO and XL were close to the end of their service life taking the fatigue cracking of the slabs under the joint influence of fatigue stress and temperature stress as the criterion. The remaining life of more than 70% of slabs in S118 HO was less than 0.1a; this proportion was more than 90% in XL. S118 TP performed relatively better. The remaining life of 45.7% slabs in S118 TP was less than 0.1a; in the other five life intervals, S118 TP had a higher proportion.

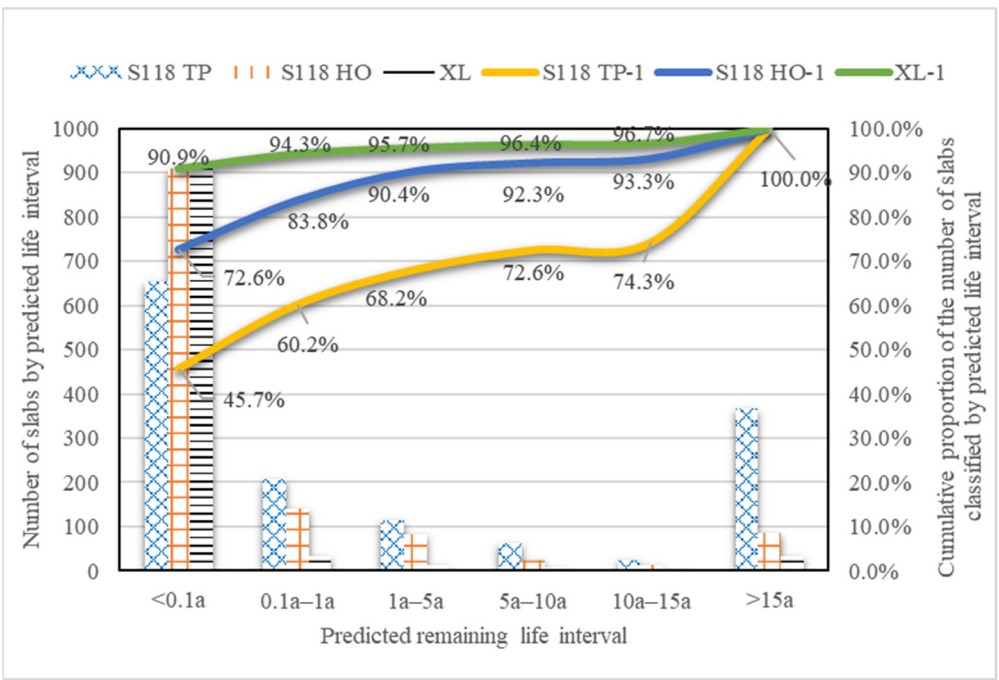

**Figure 10.** Distribution of predicted remaining life of three pavement sections in different intervals.

The mean remaining life of S118 TP, S118 HO and XL was about 2.7a, 2.2a and 2.1a respectively. Based on Table 2, the three pavement sections have been used for about 26a, 27a and 22a. Thus, their actual years of use were about 29a, 29a, and 24a. Compared to the 30-year design life of highway and first-class highway cement pavements, the difference was most pronounced for the XL section, with the S118 TP and S118 HO sections only losing 1a. This phenomenon may be related to the limestone stabilized base layer used in XL which is a kind of pavement material that is not resistant to water washing, especially in hot and rainy areas. At the same time, it may also relate to the rapid growth in traffic. The traffic volume of XL showed the largest growth rate of 7%.

In general, the distribution of the remaining life and the corresponding technical condition of the pavement may be helpful for determining the maintenance strategy of the project. This is related to the reasonable arrangement of limited maintenance funds and the maintenance priority of different sub-sections. The shorter the remaining life, the higher the maintenance treatment priority. The maintenance plan for different sub-sections can be decided based on the detection results of GPR and FWD. The logic flow of the maintenance decision-making system is shown in Figure 11.

Considering XL as an example, the investment analysis of the future maintenance plan showed that the EIRR (economic internal rate of return) was 15.5% greater than the social discount rate of 8% and the ENPV (economic net present value) was 1.05 million RMB greater than zero. The maintenance plan and maintenance investment are feasible in terms of economic benefits.

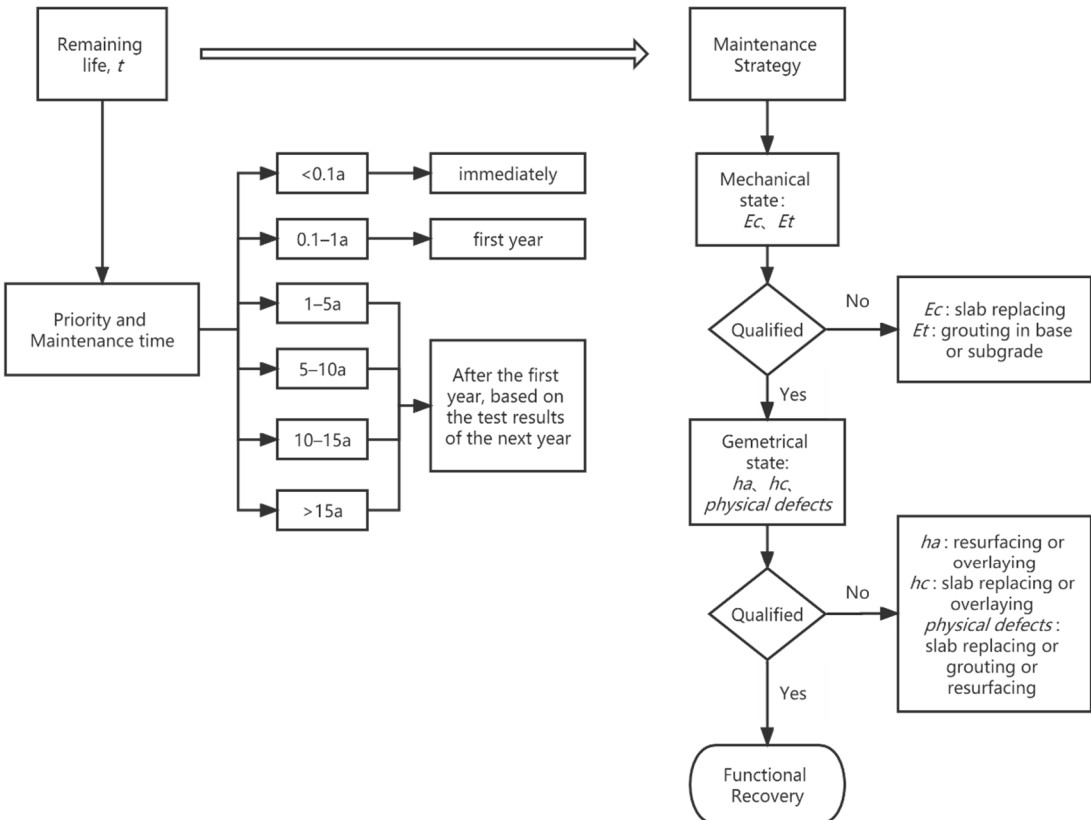

**Figure 11.** Logic flow of maintenance decision-making system.

## 5. Conclusions

The paper proposes a fast and non-destructive model for predicting the remaining life of PCC rigid pavement, with or without asphalt overlay. The prediction model was applied to three typical pavement sections. GPR was utilized to assess the geometric parameters for the predictive model and the physical state of the pavement, and FWD was utilized to assess the mechanical parameters. The main conclusions are as follows:

(1) A remaining life prediction model for rigid PCC pavement, with or without asphalt overlay, was proposed based on an inverse design concept and an elastic foundation single-layer slab model. Only four to five parameters require to be detected with the application of integrated non-destructive detection technologies on site, which greatly simplifies the life prediction process and improves the presentation of prediction results, with large amounts of data acquired quickly and non-destructively.

(2) A two- or three-layer mechanical model in the back-calculation process for rigid pavement, with or without asphalt overlay, was transferred from the multilayer (more than two or three layers) mechanical model to determine the equivalent elastic modulus of foundation underneath the concrete layer and the concrete layer bending modulus. The ratios of the back-calculated modulus and the laboratory test modulus ranged from 0.9 to 1.1, with a mean ratio of 1.01.

(3) The success rate of the prediction model when applied to three pavement sections was as high as 96%. The remaining statistically based life intervals showed good correspondence with the mechanical parameters, geometric parameters, and the physical state of the pavement. The prediction model proposed in this research has good engineering applicability and feasibility.

(4) A maintenance treatment decision-making system was proposed based on the distribution of the remaining life and the corresponding technical condition of the pavement. This was shown to be feasible in terms of economic benefits, on the basis that the pavement condition was above the minimum required standard.

In the future, we will compare our study results to those from previous research studies to evaluate the model's advantages in terms of prediction accuracy, speed, and representativeness.

**Author Contributions:** Conceptualization, C.X., J.Y. and D.W.; methodology, C.X. and Z.Q.; investigation, X.H., W.T., Z.W., W.L. and X.W.; formal analysis, W.L., X.W. and W.T.; writing—review and editing, D.W. and X.H. All authors have read and agreed to the published version of the manuscript.

**Funding:** The authors would like to acknowledge the financial support provided by the "National Natural Science Foundation of China" (52178426).

**Institutional Review Board Statement:** Not applicable.

**Informed Consent Statement:** Informed consent was obtained from all subjects involved in the study.

**Data Availability Statement:** The data presented in this study are available on request from the corresponding author.

**Acknowledgments:** The authors also want to thank the Research Project of Guangdong Provincial Communications Group on Key Technology of Pavement Balanced Durability Design and Solid Waste Recycling for its support. All the help and support are greatly appreciated.

**Conflicts of Interest:** The authors declare no conflict of interest.

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
