# Peer review of "A Fast and Non-Destructive Prediction Model for Remaining Life of Rigid Pavement with or without Asphalt Overlay"

_buildings, doi:10.3390/buildings12070868_

Round 1
Reviewer 1 Report
Accepted with minor modification.
The topic of the paper is interesting and needs to be revised. However, some comments need to be addressed:
- The title is very long. However, I suggest that it needs to be changed to reflect the manuscript.
- The motivation of study need to be shown in the introduction section?
- What makes this paper different from the others which use predicting the remaining life of rigid pavement ones?
- The authors should use this study as a comparison to previous research.
Reviewer 2 Report
The paper proposes a fast and non-destructive model for predicting remaining life of PCC (Portland cement concrete) rigid pavement with or without asphalt overlay integrating inverse design concept. The prediction model was applied in three typical pavement sections with. GPR (ground penetrating radar) was utilized for the assessing geometrical parameters for the predictive model and the physical state of the pavement, and FWD (falling weight detector) was utilized for assessing the mechanical parameters.
The topic is interesting and the paper is well-structured, nevertheless I would suggest some minor revisions to improve the paper before publications.
1) I would suggest a careful control of the language and the correction of various typos in the text.
2) the equations proposed in the model, that is the eqs. 1-21, are they present in the literature or in the standards? it would be advisable to specify their provenance and cite the sources in the literature. If they are proposed by the authors, it would be appropriate to illustrate the background from which they were derived.
3) in numerous equations there are parameters for which precise numerical values ​​are assumed. Where do these values ​​come from? why were those chosen?
4) Pg. 8, line 235. it would be appropriate to include the websites cited among the bibliographic references.
5) it would be necessary to specify the instrumentation used for the GPR and for the FWD (model, brand, device, etc.).
6) Fig. 1 (a) and Fig. 2. the writings are not very clear, it is better to enlarge them.
7) Pg. 13 lines 353-358. It would be advisable to describe the cores extraction and testing in greater detail (localization of samples, instrumentation used, reference standards, results obtained in terms of mean values ​​and standard deviations).
8) in the conclusions it would be appropriate to insert a short opening paragraph that summarizes the work done.
